# Supplementing Monosodium Glutamate in Sow Diets Enhances Reproductive Performance in Lactating Sows and Improves the Growth of Suckling Piglets

**DOI:** 10.3390/ani14121714

**Published:** 2024-06-07

**Authors:** Tian Xiang Li, In Ho Kim

**Affiliations:** Department of Animal Resource and Science, Dankook University, Cheonan 31116, Republic of Korea; tianxiangl589@gmail.com

**Keywords:** sow, neonatal growth, monosodium glutamate, lactation, milk yield, suckling piglets

## Abstract

**Simple Summary:**

At present, it is particularly important to reduce costs and improve economic efficiency in farm production. However, in intensive production, the situation of insufficient milk production in sows and the poor growth of piglets has become increasingly serious. In this context, how to increase sow milk production and improve piglet growth is the primary goal of researchers. Some studies have shown that feeding MSG to sows can enhance the growth of newborn pigs, but some researchers suspect that the addition of MSG is not safe for sows. Therefore, this experiment aims to study the impact of MSG on sows and piglets. Our research results prove that MSG is safe and effective, can reduce costs, improve economic benefits, and is suitable for intensive production.

**Abstract:**

In most current farm operations, lactating sows need to overcome reproductive and environmental stresses that have resulted in poor sow production performance and piglet growth. Therefore, this study aimed to investigate the effects of in-feed supplementation of monosodium glutamate (MSG) in sows during late gestation lactation in regard to litter performance. The study subjects were 12 multi-parity sows (Landrace × Large White), farrowing sows with an average parity of four (three with three parities, seven with four parities, and two with five parities). They were randomly divided into the following two diet groups: the basal diet as a control (CON) group based on corn and soybean meal; and the basal diet + 2% MSG group. The experimental time ranged from 109 days before delivery to 21 days after delivery. There were six sows in each group, and each sow served as the experimental unit. There were no significant differences (*p* > 0.05) in body weight (BW), back fat (BF) thickness and estrus interval between sows supplemented with 2% MSG in their diets before and after farrowing and during weaning (*p* > 0.05). However, MSG-treated sows tended to increase BW loss at farrowing more than the CON group (*p* = 0.093) but lost less weight during lactation than the CON group (*p* = 0.019). There were no significant differences in the body condition scores (BCSs) and BF loss of the two groups of sows before and after farrowing and at weaning (*p* > 0.05). There was no significant difference in the weight of newborn piglets between the two groups of sows (*p* > 0.05). The weaning weight (*p* = 0.020) and average daily gain (ADG) (*p* = 0.045) of suckling piglets were higher in the MSG treated group compared to the CON group. The daily milk production of sows in the MSG treatment group was higher compared to the CON group (*p* = 0.045). The protein concentration of milk at week 3 (*p* = 0.060) and fat concentration of milk at week 5 (*p* = 0.095) of the MSG-supplemented sows tended to increase more than the CON group. In summary, the dietary inclusion of MSG supplementation had a beneficial effect on the late gestating sows and their piglet’s growth and milk production. Our research has shown that the addition of 2% MSG in late gestation and lactation diet would be beneficial for both sow and piglet production.

## 1. Introduction

With the continuous improvement of modern artificial breeding technology, the number of litters produced by lactating sows is also increasing. This results in a greater demand for milk production from lactating sows that conflicts with the sow’s low appetite in the early postpartum period [1]. The insufficient milk production of sows directly leads to a poor growth status in newborn piglets and increases the mortality of underweight piglets [2]. In addition, lactating sows that convert maternal fat into milk lose more weight, extend the interval between estrus, and reduce pregnancy and embryo survival rates [3].

Intestinal health is the basis for overall pig health, maximum growth performance, and feed utilization efficiency [4]. However, after sow pregnancy, piglets’ feed intake is usually limited due to insufficient milk production when the sow responds to various stress factors [5]. The intestines of piglets atrophy and induce intestinal oxidative stress, and the deficiency of various nutrients, including glutamate, leads to a related growth depression syndrome in piglets [6,7].

Glutamate is the second most abundant peptide-binding amino acid (AA) in breast milk after proline [8]. As a substrate and activator of protein synthesis, glutamate has great potential to improve sow milk production [9]. At the cellular level, glutamate is physiologically required for the synthesis of proteins and other nitrogen-containing substances (including glutathione and arginine), and it has key metabolic functions in the body [10]. Therefore, this nutrient plays an important role in improving pig health, survival, growth, development, lactation, and reproduction.

Studies have shown that almost all cells of lactating sows can form glutamate from glutamine, branched-chain amino acids (BCAAs), alanine, and aspartate through different enzymes (such as phosphate-activated glutaminase, glutamine: fructose-6-phosphate) [11]. Endogenous glutamate in pigs comes from the small intestine, liver, skeletal muscle, kidneys, etc. [12]. However, in the small intestine, glutamate only serves as an intermediate product of these reactions, and the net production of glutamate from the pig’s small intestine is very small [13]. Stoll and Burrin [14] found that the glutamate synthesized in the liver and skeletal muscle is not enough to meet the muscle’s demand for glutamate. Research by Liu et al. [15] has shown that providing exogenous or endogenous glutamate can promote the rapid growth of pigs. Similarly, researchers such as Lei et al. [16] found that breast tissue can extensively utilize BCAAs, and the absorption of BCAAs is much greater than the output in the form of milk protein. Therefore, the uptake of BCAAs in the body by breast tissue is used to synthesize glutamate and glutamine [17]. This also makes up for the lack of glutamate and glutamine uptake by the mammary gland from arterial blood during lactation. In the current feeding plan, the intake of protein and BCAAs in low-protein diets is far from sufficient for the output of milk protein in high-producing sows [18]. Although lactating sows mobilize their protein reserves to provide glutamate for milk production, this ability has physiological limits, as excessive loss of protein in the body is contradictory to health or survival [19]. Therefore, lactating sows may not be able to synthesize enough glutamate to support maximum milk production. When Manso et al. [20] added feed-grade glutamine and glutamic acid mixture to the diet of primiparous sows 30 days before farrowing and 21 days after farrowing, they found that it increased the free glutamine content in the pig milk. Rezaei, Gabriel and Wu [6] added BCAAs to lactating sow feeds to increase milk production of lactating sows. Zhang et al. [21] added different doses of sodium glutamate to sow feed and found that the expression of glutamate receptors and glutamate transporters increased in the small intestine, however, this did not affect the weight of piglets when they grew to weaning.

Studies have shown that adding 0–2% MSG to the diet of weaned piglets does not affect feed intake, but the feed intake of pigs with 4% added MSG was 15% lower than that of pigs without MSG [22]. Similarly, another study showed that adding 1–2% MSG to the diet of lactating sows can increase the sow’s milk production but has no effect on the sow’s feed intake [23]. Although different researchers have mixed opinions on the use of MSG, a large number of research results show that the use of MSG is safe and effective.

At present, little is known about the impact of AAs on sow milk production during lactation. Therefore, this experiment aimed to verify that adding MSG to the diet may increase sow milk production and improve piglet growth. We hypothesized that adding 2% MSG to the basal diet of lactating sows would increase sow milk production and improve piglet growth.

## 2. Materials and Methods

### 2.1. Animal Ethics

All animal procedures used in this experiment were reviewed and approved (DK-2-2301) by the Institutional Animal Care and Use Committee of Dankook University (Cheonan, Republic of Korea).

### 2.2. Animals Housing and Management for Feeding Trials

An amount of 12 multi-parity sows (Landrace × Large White), farrowing sows with an average parity of 4 (3 with 3 parities, 7 with 4 parities, and 2 with 5 parities) were randomly divided into the following two dietary groups: basal diet as control (feed based on corn and soybean meal); basal diet + 2% MSG. Twelve sows were housed individually in 1.8 × 2.5 m pens with plastic-coated perforated floors and fed 3.5 kg of corn and soybean meal diet throughout the gestation. The experimental diet (Table 1), a lactation diet that meets the requirements of [24], was fed one week before and three weeks after parturition. The diet adopts the method of isonitrogen and isocaloric, wherein an appropriate amount of L-alanine is added to the diet to make the nitrogen content of the control group and the treatment group the same. Additionally, 0% MSG was added to the control group, while 2% MSG (experimental group) was added to the experimental diet. The diet was fed twice and drinking water was available as ad libitum. The feed intake of the sows was recorded every day, and the average daily feed intake (ADFI) was calculated on the 7th and 28th days of the experiment. On the day of farrowing (day 7 of the experiment), in order to reduce variation within the litter, newborn piglets were cross-fed. The number of piglets per sow was standardized to 9. At farrowing (the farrowing day was regarded as the 0 day of lactation), live piglets and stillborn piglets were recorded (deformed piglets were regarded as stillborn), and the birth weights of live piglets were measured, respectively. The piglets received plastic ear tags and iron supplementation when they were 3 days old, and the piglets were subjected to routine treatments such as tail bobbing, teeth clipping, and castration. The piglets were kept in an incubator at 22–32 °C, and the temperature was controlled with an auxiliary heating lamp. They were fed 3 times a day, starting from 2 kg on the first day of lactation and gradually increasing to 1.0 kg per day until the 6th day (Figure 1). The sows could then eat ad libitum until the 21st day of lactation. During lactation, the feed intake of individual sows was recorded daily. All pigs had free access to water. During the entire experiment, the piglets did not receive the sow’s feed, nor did the piglets receive any creep feed with liquid feed instead of milk (Figure 2).

### 2.3. Measurement

The BF thickness and BW of fasting sows were measured on day 108th, the day after farrowing, and the 21st day after birth. BF thickness was measured with an ultrasonic instrument at the left dorsal midline of the 10th rib of the sow (Renco Lean-Meater, Hennepin County, MN, USA).

BCS assessment was performed on day 108 of gestation, 24 h postpartum, and day 21 postpartum using the 5-point method of Knauer and Baitinger [25], focusing on the sow’s hip joint and spine. The score evaluation is as follows: 1 = looks very thin; 2 = the palm can feel the bones are hard and there is no pressure; 3 = the palm can be sunken when touched; 4 = the hand cannot distinguish the shape of the bones; 5 = obesity.

After weaning, 12 sows were transferred to the gestation pen, and the estrus interval of the sows was recorded. A sow is considered to be in estrus when she stands and responds to the back pressure test while accompanied by a boar. Estrus identification is carried out at 10:00 and 15:00 every day, and the period from weaning to estrus is the estrus interval.

### 2.4. Collection of Milk from Sows

To obtain sow milk samples, 20 IU oxytocin was injected through the ear vein of the sow to induce milk flow into the teat duct, and then the milk was manually expressed [26]. Milk samples (50 mL/mammary gland) were collected from three different mammary glands (anterior, middle, and posterior). Equal amounts of milk (20 mL) from each gland of the sow were mixed and the samples were stored at −20 °C for subsequent analysis.

### 2.5. Assay for Colostrum and Milk Composition

Thawed colostrum and milk samples were analyzed using a mammoscope FTIR automatic milk analyzer (Delta, Flevoland, Zeewolde, The Netherlands) to assess the fat, protein, and lactose content in the milk. The result is calculated as a percentage of colostrum and milk. The evaluation of milk production during lactation is based on the ADG of piglets and the number of litters. The calculation formula is as follows [27]: milk production = ADG of piglets × number of litters ×Days of lactation × 4. This gives the average milk production of the sows.

### 2.6. Statistical Analysis

All data in this experiment were analyzed using SAS 9.4 software (SAS Institute, Cary, NC, USA). A non-parametrical Mann−Whitney U test was used to determine the significance of differences between treatments to examine the response to the addition of MSG in the basal diet. The standard error of the mean (SEM) is a way of expressing the variability of data, using pens as the unit of testing. *p* < 0.05 is significance and *p* < 0.1 is trend.

## 3. Results

### 3.1. Reproduction Performance of Sows

The reproductive indicators of sows are shown in Table 2. MSG-treated sows tended to increase BW loss at farrowing more than the CON group (*p* = 0.093) but lost less weight during lactation than the CON group (*p* = 0.019).

There were no significant changes in the total number of births, the number of stillbirths, the number of survivors, and the survival rate of piglets in the control group and the diet treatment group (*p* > 0.05). There were no statistical differences in the interval from weaning to estrus, BF thickness, BF loss, feed intake, and the BCSs of sows in each group before and after delivery (*p* > 0.05).

### 3.2. Growth Performance of Suckling Piglets

The growth performance of suckling piglets is shown in Table 3: The BW of suckling piglets in the MSG-added diet was statistically significant (*p* = 0.020) in comparison to the CON group. In addition, the ADG of suckling piglets in the dietary treatment group also increased significantly throughout the lactation period (*p* = 0.045).

### 3.3. Milk Profile in Lactating Sow

As shown in Table 4, during the entire lactation period, the milk production of sows in the MSG-added diet treatment group was significantly increased compared to the CON group (*p* = 0.045). In the second week of lactation, the milk protein content of sows in the MSG-added group tended to increase compared to the CON group (*p* < 0.10). In the third week of lactation, the fat concentration in the milk of sows supplemented with the MSG diet tended to increase compared to the CON group (*p* < 0.10) (Figure 3).

## 4. Discussion

Due to the improvement of modern artificial breeding technology, modern high-yield sows can ovulate 20–30 oocytes but can only give birth to 10–15 live piglets at term [28]. The number of pigs born alive is limited by placental development and naturally occurring intrauterine growth [29]. Studies have shown that glutamic acid is the main substrate for the synthesis of arginine in most mammals (including pigs) [30]. Research by Wu et al. [31] and others has shown that arginine is important in pig nutrition and physiology. The reason is that polyamines and nitric oxide (a product of arginine) affect mammalian placental angiogenesis and growth [32]. Therefore, insufficient glutamate content in the diet will affect fetal survival and growth. Studies have showed that adding MSG to the basal diet of lactating sows can improve the survival rate of piglets [33]. In this study, adding 2% MSG after day 107 of gestation had no effect on the number of newborn piglets, whether they were live births or stillbirths. Sows fed MSG had the same number of stillbirths as sows in the control group. Dystocia and lack of motivation in sows during the late farrowing period are all causes of stillbirth [34]. Lack of supervision is also an important factor leading to stillbirth [35]. In this case, because MSG was added too late, it was difficult to determine the cause of the stillbirth. Since there was no difference in the total number of births between groups, the number of stillbirths was the same. Therefore, there was no difference in the number of live births between groups.

Studies have shown that the weight loss of lactating sows during farrowing is restricted by the sow’s nutritional level, feed intake and piglet birth weight [36]. This study was conducted under the condition that the same nutritional level of the diet was provided and that the feed intake of the sows in the week before farrowing was the same and restricted. Therefore, the main factor affecting the weight loss of lactating sows during farrowing is the weight of piglets at birth. In our study, the average farrowing weight of sows supplemented with MSG is similar to that of the CON group. It is worth noting that, at the beginning of the experiment, the average body weight of the lactating sows supplemented with MSG was lower than that of the control group. However, the piglets produced by the two groups were of similar weight. It remains to be determined whether glutamate in milk is sufficient to achieve maximum growth and development in piglets raised from sows, especially those with low birth weights. Therefore, we speculate that the reason for the weight loss trend of sows during farrowing may be that, after the addition of MSG, lactating sows with smaller body weights will increase their protein synthesis under the action of MSG. The utilization rate of dietary nutrients and the utilization rate of arginine can ensure the normal development of piglets. Although sows lost more weight at farrowing, the BCSs did not differ significantly between groups. The Hou and Wu [37] study showed that, in pig tissues, glutamic acid provides an amino group for the conversion of 4-hydroxyproline (a product of collagen degradation) into glycine (a nutritionally essential amino acid for piglets). Therefore, a large amount of glutamate needs to be provided in the uterus during pregnancy. Therefore, during pregnancy, most of the sodium glutamate supplemented to lactating sows is used to ensure the normal intrauterine development of piglets. Similarly, Wu [38] found that arginine and glutamine can activate the mechanistic target of the rapamycin cell signaling pathway, thereby stimulating protein and fat synthesis in skeletal muscles and other tissues of lactating sows and inhibiting proteolysis and fat. This mechanism also explains why the sows that added 2% sodium glutamate in this experiment had less BF loss during lactation. It also shows that adding sodium glutamate to feed can reduce the weight loss of lactating sows.

Adding 2% MSG to the sow’s basal diet one week before farrowing and during lactation can increase the sow’s milk production as well as the tendency for the protein and fat content in the milk to increase.

Insufficient milk production of sows limits the growth performance of newborn piglets [3]. However, there are currently not many studies showing that insufficient milk production in sows is due to a lack of all AAs or certain AAs. Li et al. [39] demonstrated that mammary gland cells of lactating sows can generate glutamate from BCAAs and a-ketoglutarate in their mitochondria and cytoplasm. This is followed by the conversion of glutamate to glutamine in the cytoplasm by glutamine synthetase. However, the lactating glands of sows lack glutaminase and proline oxidase and cannot hydrolyze glutamine to glutamate or form glutamate from proline [40]. Therefore, glutamate in the mammary gland only comes from maternal blood and the synthesis of BCAAs. Research by Manjarin, Bequette, Wu and Trottier [3] showed that the uptake of AAs by sow lactation glands is positively correlated with the concentration of AAs in arterial blood. In addition, research by Boutry et al. [41] showed that glutamic acid can inhibit the degradation of BCAAs by porcine intestinal cells. Secondly, glutamic acid can reduce the number of *Klebsiella* in the mixed flora of a pig’s small intestine [42], and *Klebsiella* is the main bacterium that degrades BCAAs, glutamine and tryptophan [43]. However, we did not observe intestinal cells, which is a limitation of this study. More research is needed in the future to prove this. The results of Rezaei, Gabriel and Wu [23] showed that adding 2% MSG to the diet of sows can increase the concentration of glutamate in plasma. The results showed that the average glutamate concentration of sows without MSG during lactation was 90 mmol/L, while the glutamate concentration of sows with glutamate added was 163 mmol/L. Precise data on the extraction of glutamate and other BCAAs from blood by the mammary gland of lactating sows are lacking. Therefore, we speculate that adding 2% MSG to sows’ diets in this experiment increased the concentrations of glutamate and BCAAs in maternal blood, thus increasing the availability of mammary gland cells to synthesize glutamine, alanine, aspartate, asparagine and protein. This also just illustrates the trend of increasing protein content in the milk of sows during the lactation period in this experiment. In our study, adding 2% sodium glutamate can tendency increase for the fat concentration in sow milk. Since glutamate, glutamine, alanine, aspartic acid, proline and arginine are substrates for gluconeogenesis in sows [44], these AAs can be utilized by white adipose tissue to synthesize fatty acids [45]. The uptake of fatty acids by sow lactation glands increases the synthesis of triglycerides in mammary epithelial cells. Overall, increases in these AAs promote breast milk synthesis and fat concentration.

The weight gain and growth of piglets are closely related to the amount and nutritional content of sow milk [23]. This study shows that adding 2% MSG to sow diets can increase milk production and milk fat content during lactation. Therefore, the results of this experiment show that piglets supplemented with sodium glutamate have greater BWs and can increase the BW and ADG of piglets during lactation, effectively improving the growth of piglets.

## 5. Conclusions

In summary, adding 2% MSG to corn and soybean meal-based diets can increase sows’ milk production by reducing BF loss during lactation. It promotes the growth of suckling piglets and improves the growth conditions of piglets. Since glutamate can have many beneficial effects on sows during lactation and improve piglet growth, sodium glutamate can be safely and effectively adapted to today’s intensive production. Therefore, it is necessary to supplement MSG during the lactation period of lactating sows.

## Figures and Tables

**Figure 1 animals-14-01714-f001:**
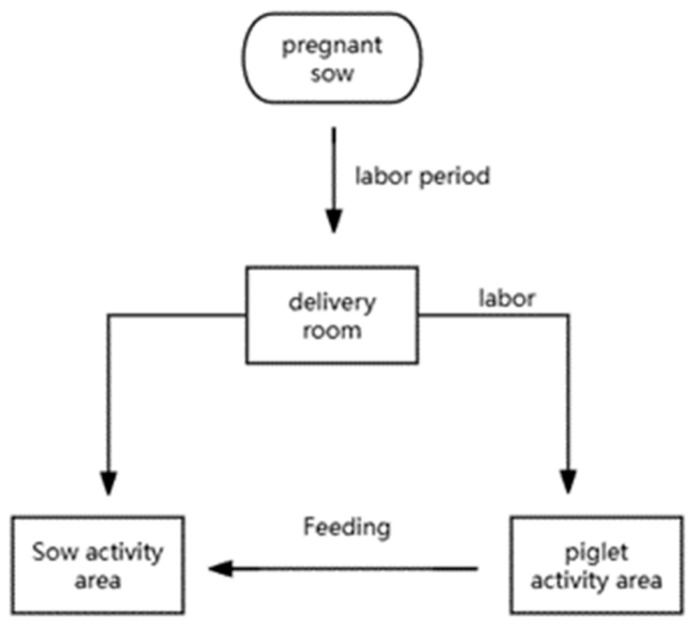
Schematic diagram of childbirth and activities.

**Figure 2 animals-14-01714-f002:**
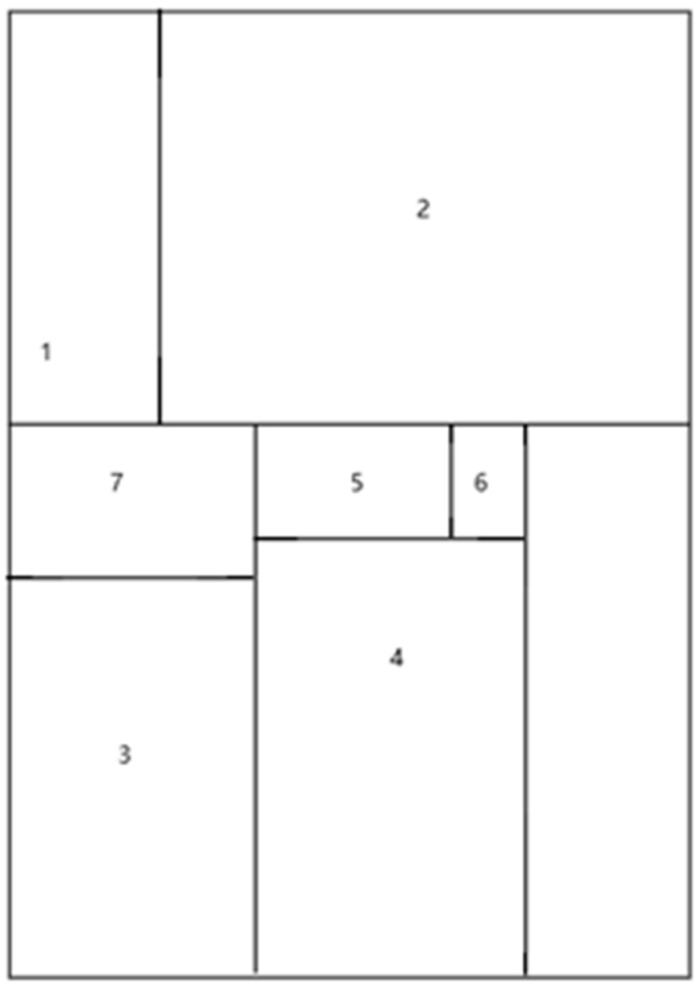
Plane figure of sow and piglet living. 1. Piglet feeding trough. 2. Piglet insulation area. 3. Piglet activity area. 4. Sow restriction area. 5. Sow trough. 6. Sow automatic waterer. 7. Piglet automatic waterer.

**Figure 3 animals-14-01714-f003:**
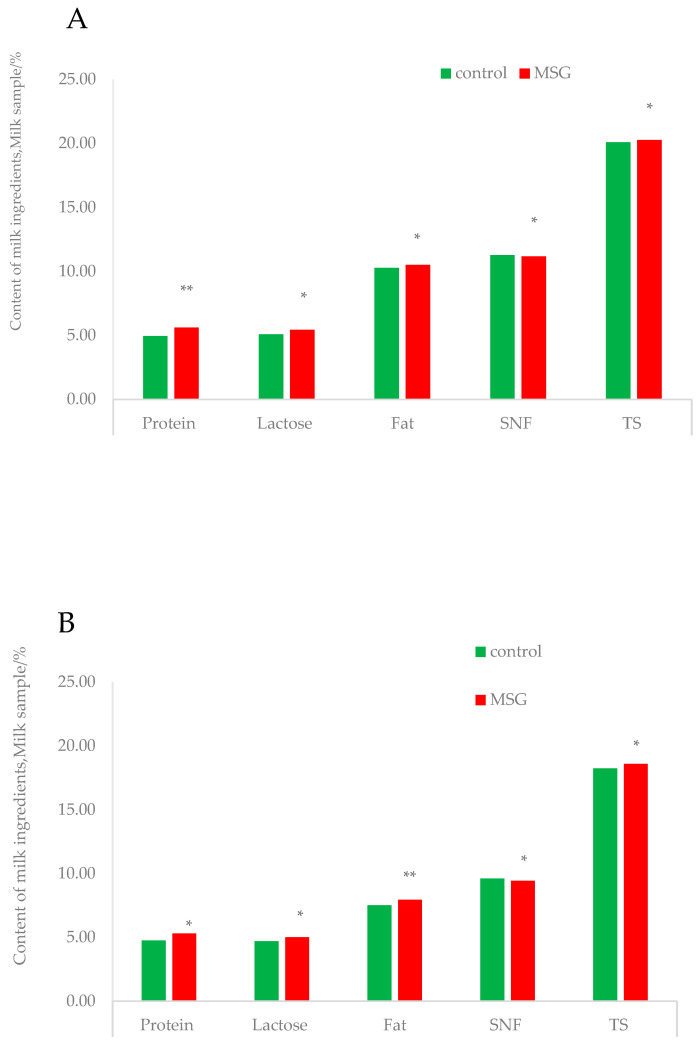
Composition of the sows’ milk between the different treatments. (**A**), Milk composition of sows in CON group and MSG group in week 2 of the experiment; (**B**), Milk composition of sows in CON group and MSG group in week 3 of the experiment. * *p* > 0.05, ** 0.05 < *p* < 0.1.

**Table 1 animals-14-01714-t001:** Composition and chemical analysis of sow diets ^1^.

Items	Basal Sow Diet	Basal Sow Diet with 2% MSG
Ingredient, % DM basis		
Corn	41.93	40.96
Wheat	23.00	23.00
Wheat bran	8.31	8.32
Soybean meal, (48% CP)	4.48	3.48
Dehulled Soybean meal, (44% CP)	12.96	11.25
Molasses	2.00	2.00
Soybean oil	3.40	4.83
Monocalcium phosphate	1.20	1.30
Limestone	1.18	1.16
MgO	0.02	0.02
Salt	0.50	0.50
L-Threonine (99%)	0.17	0.22
DL-Methionine (99%)	0.02	0.03
L-Lysine HCl, (78%)	0.31	0.41
Vit/Mineral premix ^2^	0.40	0.40
Choline (25%)	0.12	0.12
Glumatic acid	-	2.00
Total	100.00	100.00
Chemical analysis (%)		
Crude protein	16.50	16.50
Metabolizable energy (kcal/kg)	3300	3300
Ether extract	5.71	7.06
Ca	0.76	0.76
Total phosphorus	0.65	0.65
Available phosphorus	0.35	0.35
L-Lysine HCl	0.96	0.96
L-Threonine	0.26	0.26
DL-Methionine	0.65	0.65
L-glutamine	3.50	3.89
L-glycine	0.77	1.52

^1^ Abbreviation: MSG, monosodium glutamate. ^2^ Provided per kg of complete diet: 16,800 IU vitamin A; 2400 IU vitamin D_3_; 108 mg vitamin E; 7.2 mg vitamin K; 18 mg Riboflavin; 80.4 mg Niacin; 2.64 mg Thiamine; 45.6 mg D-Pantothenic; 0.06 mg. Cobalamine; 12 mg Cu (as CuSO_4_); 60 mg Zn (as ZnSO_4_); 24 mg Mn (as MnSO_4_); 0.6 mg I (as Ca(IO_3_)_2_); 0.36 mg Se (as Na_2_SeO_3_).

**Table 2 animals-14-01714-t002:** The effect of dietary MSG supplementation on reproduction performance in lactating sows ^1^.

Items	CON	MSG	SEM ^2^	*p*-Value
Parity	3.5	3.2	0.5	0.788
Litter size				
Total birth, head	12.0	12.7	1.3	0.686
Mummification, head	0.2	0.0	0.2	0.405
Stillbirth, head	0.3	0.3	0.3	1.0
Total alive, head	11.5	12.3	1.3	0.682
SUR1 ^3^, %	95.85	97.70	2.6	0.530
Body weight, kg				
Initial	225.1	208.7	19.4	0.379
Farrowing	206.0	187.9	19.6	0.379
Weaning	190.4	173.0	18.6	0.378
Ovulation	192.8	175.9	13.4	0.379
Body weight difference 1 ^4^	19.1	20.8	0.8	0.093
Body weight difference 2 ^4^	15.7 ^b^	14.8 ^a^	0.4	0.019
Body weight difference 3 ^4^	2.4	2.9	0.8	0.520
Backfat thickness, mm				
Initial	19.8	19.0	1.0	0.417
Farrowing	18.2	17.8	0.9	0.557
Weaning	15.8	15.8	0.8	1.0
Ovulation	16.7	16.8	0.8	1.0
Backfat thickness difference 1 ^5^	1.7	1.2	0.4	0.112
Backfat thickness difference 2 ^5^	2.3	2.0	0.4	0.387
Backfat thickness difference 3 ^5^	0.8	1.0	0.6	0.798
Body condition score				
Initial	3.5	3.0	0.4	0.153
Farrowing	3.0	2.8	0.3	0.341
Weaning	2.6	2.7	0.3	0.784
Ovulation	3.1	3.2	0.4	0.738
ADFI, kg				
Pregnant	3.50	3.50	-	-
Lactation	7.10	7.20	0.16	0.422
Ovulation	4.00	4.00	-	-
Estrus interval, d	5.2	4.7	1.2	0.678

^a,b^ in superscripts in the same row describe statistically significant differences (*p* < 0.05) between mean values. ^1^ Abbreviation: CON, basal diet; MSG, basal diet + 2% monosodium glutamate. ^2^ Standard error of means. ^3^ SUR1: survival rate of number of alive pig per number of total born pigs. ^4^ Body weight difference: Initial to Farrowing; 2, Farrowing to Weaning; 3, Weaning to Ovulation. ^5^ Backfat thickness difference: Initial to Farrowing; 2, Farrowing to Weaning; 3, Weaning to Ovulation. ADFI, average daily feed intake.

**Table 3 animals-14-01714-t003:** The effect of dietary MSG supplementation on growth performance in suckling piglets ^1^.

Items	CON	MSG	SEM ^2^	*p*-Value
INO	11.5	11.8	0.4	0.282
FNO	11.3	11.7	0.5	0.523
SUR2 ^3^, %	98.49	98.61	2.11	1.0
Body weight, kg				
Birth weight	1.25	1.26	0.11	0.873
Weaning	6.02 ^b^	6.43 ^a^	0.19	0.020
ADG, g	228 ^b^	246 ^a^	7.8	0.045

^a,b^ in superscripts in the same row describe statistically significant differences (*p* < 0.05) between mean values. ^1^ Abbreviation: CON, basal diet; MSG, basal diet + 2% monosodium glutamate. ^2^ Standard error of means. ^3^ survival rate during lactation. INO, the number of initial suckling piglet; FNO, the number of finish suckling piglet. ADG, average daily gain.

**Table 4 animals-14-01714-t004:** The effect of dietary MSG supplementation on milk profile in lactating sows ^1^.

Items	CON	MSG	SEM ^2^	*p*-Value
Milk yield, kg	10.465 ^b^	11.659 ^a^	0.50	0.045
Week 2				
Fat, %	10.28	10.50	0.3827	0.600
Protein, %	4.95	5.62	0.3125	0.060
Lactose, %	5.09	5.43	0.3782	0.296
Solids Not Fat, %	11.28	11.17	0.4738	0.401
Total-solids, %	20.10	20.26	0.2723	0.676
Frozen Point, ℃	−0.65	−0.65	-	-
Week 3				
Fat, %	7.52	7.95	0.1765	0.095
Protein, %	4.75	5.30	0.332	0.144
Lactose, %	4.70	5.02	0.2978	0.402
Solids Not Fat, %	9.61	9.44	0.5753	0.835
Total-solids, %	18.25	18.60	0.28	0.173
Frozen Point, ℃	−0.58	−0.58	-	-

^a,b^ in superscripts in the same row describe statistically significant differences (*p* < 0.05) between mean values. ^1^ Abbreviation: CON, basal diet; MSG, basal diet + 2% monosodium glutamate. ^2^ Standard error of means.

## Data Availability

The data presented in this study are available on request from the corresponding author.

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
