# Peer review of "Supplementing Monosodium Glutamate in Sow Diets Enhances Reproductive Performance in Lactating Sows and Improves the Growth of Suckling Piglets"

_animals, 2024, doi:10.3390/ani14121714_

Round 1

Reviewer 1 Report

Comments and Suggestions for Authors

The present research evaluates the dietary supplementation of pregnant and lactating females with MSG and its effect on reproductive parameters of the sows and productive parameters of the sow and its litters.

1- Because the study incorporates several measurements on sows and their litters, it would be useful to make a schematic figure of the main activities and evaluations on sows and piglets. This complements the understanding of experimental design within the material and methods section.

2- The tables are very useful to show productive results of animals exposed to tests. However, other forms of visualization of results could be incorporated, such as the use of graphs to detail the composition of the sows' milk between the different treatments. This change could substantially improve the current quality of the article.

3- The discussion presented in this research is precise and comprehensive, providing a clear response to the results obtained. An important point is that the conclusion mentions the feasibility of applying the study to current intensive pig production systems.

4- The writing and errors in the writing should be reviewed and improved. for example:

Line 28: remove capitalization add space to keywords

Line 92: add space in “(DK-2-91 2301)by”

Line 104: add space in “Additionally,0%”

Line 109: can you explain the “cross-fed of newborn piglets

Line 206: add space in “piglets.In”

5- Table 1: unify nomenclature of diets. It is not clear if the experimental diet is called TRT or basal diet + glutamic acid 2% or glutamic acid 2%

6- Tables 2-3:  A group of sows "TRT1" has not been defined previously. Is there a group TRT2?

Author Response

请参阅附件!

Reviewer 2 Report

Comments and Suggestions for Authors

Supplementing monosodium glutamate in the sow diets from day 109 of gestation to the entire lactation period enhances milk production and milk fat concentration in lactating sows and improves the growth of suckling piglets

Dear Authors,

The manuscript is interesting, and quite well prepared, but there are some corrections required from statistical, nutritional and editorial point of view.

Below I add some suggestions helpful in this process:

Line 23

In text of manuscript is described that protein concentration of milk at week 3 of sows in MSG treatment was higher (p=0.0597) compared to control. In this case only tendency can be confirmed (higher than critical value p=0.05). Daily milk production and fat milk concentration was significantly different as described.

Line 34

Ordinal numbers, as was described in Instructions for Authors, must be used instead of Authors, Teams. Sometimes it is problematic, but it works.

Line 95

In text is calving sows, but in my opinion pregnant or farrowing sows will better fit.

Additionally, parities there are second experimental factor because it will be difficult to obtain 2 groups with identical number of sows from different parities numbers (homogenic treatments). Maybe non-parametrical U-Mann-Whitney test could be useful here?

Line 104

‘…Additionally,0%MSG…’ (space required).

Line 118

‘…until the 21st day of lactation…’ , 21st day can be used (st in superscript).

Line 121

First column, ingredients.

Soybean meal have described percentage value 48%,  but information about crude protein is required (48 % CP). The same situation with dehulled soybean meal (…%).

Amino acids must be describe more accurate:

L-Threonine (99%)

DL-Methionine (99%)

L-Lysine HCl (78%)

Second part of table: chemical composition, can have added information that values was calculated, but there is no information added about ingredient analysis in materials and methods. Chemical analysis of feed ingredients must be described, what can allow to calculate chemical composition of entire diet.

Also Metabolizable energy and nutrients must be describe more precise:

Crude protein (CP)

Metabolizable energy (ME)

In text is FAT, but maybe ether extract should be emphasized (EE)?

P, %. Phosphorus (total or available?, must be precise determined)

Amino acids: L-lysine, L-threonine, DL-methionine, L-glutamine, L-glycine

Line 125

If: P, LYS, MET was added in Table 1, there are not necessarily.

Line 158

Information about test checking normality of distribution of data is required, but description in line 95 suggest lack of homogeneity/ equality of sows in the same number of parity. Maybe better for whole experiment conduct non-parametrical U-Mann-Whitney test.

Line 168

Table 2

Letters a, b in superscripts are required to described significance of differences.

Items

CON

TRT1

SEM2

p-value

Body weight difference 1

19.1b

20.8a

0.7

0.044

Body weight difference 2

15.7a

14.8b

0.3

0.021

Letters a, b in superscripts in the same row describe statistically significant  difference (p<0.05) between mean values.

Line 182

Table 3

Title of table must be adapted to margin size.

p-value instead of P-value must be used in heading.

Letters a, b in superscripts are required to described significance of differences, with the same description below the table.

Line 192

Table 4

Title of table must be adapted to margin size.

p-value instead of P-value must be used in heading.

Letters a, b in superscripts are required to described significance of differences, with the same description below the table.

Line 300

References

Dots/points in Journal’s name abbreviations must be used.

Year of publication must be emphasized by bold using, and must be present after Journal name (ie.

Hou, Y.; Wu, G. L-Glutamate nutrition and metabolism in swine. Amino acids 2018, 50(11), 1497-1510. doi:10.1007/s00726-018-2634-3.

All Authors must be presented in each reference. Without shortcut in form: et al. or ‘…’

Reviewer 3 Report

Comments and Suggestions for Authors

Dear Authors,

The manuscript "Supplementing monosodium glutamate in the sow diets from day 109 of gestation to the entire lactation period enhances milk production and milk fat concentration in lactating sows and improves the growth of suckling piglets" by Tian Xiang Li and In Ho Kim (animals-3030930) is devoted to an important problem of the animal husbandry. The aim of this study – “to investigate the effects of in-feed supplementation of monosodium glutamate (MSG) in sows (Landrace × Yorkshire) during late gestation a lactation on litter performance”. The authors found that an inclusion of MSG supplementation in the sow diet had “a beneficial effect on the late gestating sows and their piglet’s growth and milk production”. Finally, the authors proposed that “the addition of 2 % MSG in late gestation and lactation diet would be beneficial for both sow and piglet production”.    

The manuscript analyze the literature works in detail and at high level of discussion. I do not doubt the technical quality of the work and feel that there is a sufficient impact on a broader readership to justify publication in the "Animals". This topic is in frame of the journal scopes; the subject matter is treated in depth. Thus, the present manuscript is actual and important, especially in the field of the animal husbandry.

There are some comments:

1. Page 3, lines 86-87.  The authors suggested (hypothesized) that “adding 2% MSG to the basal diet of lactating sows would increase sow milk production and improve piglet growth”. It is important to explain here (in the part 1. Introduction) or in the part 4. (Discussion) why the authors adding only 2% MSG (not 3%, 4%, etc.) to the sow’s basal diet. Page 7, lines 224-225. In the part “4. Discussion” the authors wrote: “There are reports  that adding 0.5% MSG to the basic diet can increase the feed intake of suckling pigs after  farrowing, but does not affect the weight at weaning (He & Wu, 2022).”

2. Page 3, lines 121-122. Table 1. “Ingredient composition of experimental diets as-fed basis”. I propose to use in the column 2) “Basal sow diet” instead of the “Basal diet” and in the column 3) “Basal sow diet with 2 % MSG” instead of the “Glumatic acid 2%”. This will be more clear to the readers.  In this case the line “Sow diet” is not necessary to use. Please, correct the chemical formula “CuSO4” instead of the “CuSO4”, as well as the following chemical formulae in the lines 124-125.

3. Page 8, lines 263-267. In the part “4. Discussion” the authors wrote: “Precise data on the extraction of glutamate and other BCAAs from blood by the mammary gland of lactating sows are lacking. Therefore, we speculate that adding 2% MSG to sows' diets in this experiment increased the concentrations of glutamate and BCAAs in maternal blood thus increasing the availability of mammary gland cells to synthesize glutamine, alanine, aspartate, asparagine and protein”. It will be great, if the authors can provide own or literature data on the glutamate content in the blood of the lactating sows.

4. Minor editing of English language required.

Comments on the Quality of English Language

Minor editing of English language required.

Reviewer 4 Report

Comments and Suggestions for Authors

The manuscript titled “Supplementing monosodium glutamate in the sow diets from day 109 of gestation to the entire lactation period enhances milk production and milk fat concentration in lactating sows and improves the growth of suckling piglets” has studied the effects of dietary MSG supplementation on sow reproduction and piglets production, and showed that it is beneficial. The study is solid and well-written. However, some concerns need to be addressed.

1. Please shorten the title, it describes the study in detail even better than the abstract.

2. The abstract would benefit from one or two sentences of background description and a bit more description of experimental design.

3. Line 11, “were blocked according to parity (4)”, was that a reference? If so, please remove it.

4. Please revise the outlook of all tables. Taking table 2 for example, it includes different categories “Parity; Body weight, kg; Body condition score; ADFI, kg” etc. with sub-items, which can be highlighted for the eye. For example, bold the text, or sth.

5. One thing missing was the study and/or discussion about the piglet intestine. As mentioned in introduction, it is important. The authors could try to either present results that they have on intestine in the current study, or discuss why it is not studied, the limitation etc.

Comments on the Quality of English Language

The English is fine, the use of punctuation can be improved. For example, P = 0.05, with space in between.

Round 2

Reviewer 2 Report

Comments and Suggestions for Authors

Dear Authors,

Thank You for revision proces:

This time only several suggestions from my side:

            1.      Simple summary must be added on the beginning of manuscript.

2.      Space in line 84,  86 and 225 before references is required, respectively [22], [23] and [33].

3.      References must be adapted to Instructions for authors:

·        Palatino linotype font required

·        Abbreviations of Journals must be used,

ie. No. 2: Journal of Animal Science, must be presented as J. Anim. Sci.

No. 4: Frontiers in Immunology, abbreviation: Front. Immunol.

1